# Metagenomic analyses of 7000 to 5500 years old coprolites excavated from the Torihama shell-mound site in the Japanese archipelago

Luca Nishimura[1,2], Akio Tanino[3], Mayumi Ajimoto[4], Takafumi Katsumura[5], Motoyuki Ogawa[5], Kae Koganebuchi[6], Daisuke Waku[7], Masahiko Kumagai[8], Ryota Sugimoto[2], Hirofumi Nakaoka[9], Hiroki Oota[6], Ituro Inoue[1,2]*

1 Human Genetics Laboratory, National Institute of Genetics, Mishima, Shizuoka, Japan, 2 Department of Genetics, School of Life Science, The Graduate University for Advanced Studies (SOKENDAI), Mishima, Shizuoka, Japan, 3 Kitasato University Graduate School of Medical Sciences, Sagamihara, Kanagawa, Japan, 4 Wakasa History Museum, Obama, Fukui, Japan, 5 Department of Anatomy, Kitasato University School of Medicine, Sagamihara, Kanagawa, Japan, 6 Department of Biological Sciences, Graduate School of Science, University of Tokyo, Tokyo, Japan, 7 Department of International Agricultural Development, Faculty of International Agriculture and Food Studies, Tokyo University of Agriculture, Tokyo, Japan, 8 Research Center for Advanced Analysis, National Agriculture and Food Research Organization, Tsukuba, Ibaraki, Japan, 9 Department of Cancer Genome Research, Sasaki Institute, Tokyo, Japan

* itinoue@nig.ac.jp

**Data Availability Statement:** All raw sequenced data are stored on DDBJ Sequence Read Archive (DRA) under the BioProject ID PRJDB15258.

## Abstract

Coprolites contain various kinds of ancient DNAs derived from gut micro-organisms, viruses, and foods, which can help to determine the gut environment of ancient peoples. Their genomic information should be helpful in elucidating the interaction between hosts and microbes for thousands of years, as well as characterizing the dietary behaviors of ancient people. We performed shotgun metagenomic sequencing on four coprolites excavated from the Torihama shell-mound site in the Japanese archipelago. The coprolites were found in the layers of the Early Jomon period, corresponding stratigraphically to 7000 to 5500 years ago. After shotgun sequencing, we found that a significant number of reads showed homology with known gut microbe, viruses, and food genomes typically found in the feces of modern humans. We detected reads derived from several types of phages and their host bacteria simultaneously, suggesting the coexistence of viruses and their hosts. The food genomes provide biological evidence for the dietary behavior of the Jomon people, consistent with previous archaeological findings. These results indicate that ancient genomic analysis of coprolites is useful for understanding the gut environment and lifestyle of ancient peoples.

## Introduction

Fossils and bones can provide insights into the physical characteristics and habitat of ancient organisms from over tens of thousands of years ago. Moreover, genome information from such remains could be available if the endogenous DNAs are in a good state of preservation.

**Funding:** This work was supported by JSPS KAKENHI (grant numbers JP17H03738(HO), JP18H05506(II), JP20H01370(HO), JP20K21405 (II), JP21H05362(HO), JP21J22509 (LN), JP21K19289(HO), JP22F22075(HO)), AMED under Grant Number JP23ek0109650h0001(II)), and Research Organization of Information and Systems (Investment program for futuristic research projects) (RS). The funders had no role in study design, data collection and analysis, decision to publish, or preparation of the manuscript.

**Competing interests:** The authors have declared that no competing interests exist.

Ancient DNA analysis has recently developed into the field of ancient genomics with the development of massive sequencing technologies. Ancient genomic studies have offered important information, such as the lineages, migration, and history of human populations [1–6]. Biological remains, such as dental stones and coprolites, contain DNAs derived from the intraoral and enteral microbes of ancient humans and/or archaic hominins [7, 8]. As the microbes in the body have been highlighted as playing roles in the health and disease of individuals, analyses of ancient microbial genomes could provide indicators of lifestyles, such as hunter–gathering or farming [9], and pathogenic states, such having Hepatitis B viruses, *Yersinia pestis*, or others [10–12].

Previously, Nishimura et al. (2021) have reported the ancient viral sequences found in the dental pulps of Jomon individuals [13]. The Jomon people were inhabitants in the Japanese archipelago about 16,000 to 2900 years ago [14]. In particular, a complete sequence of sipho-virus contig89 was reported from dental pulp, providing the evolutionary process of the virus. To comprehend the microbiomes of the Jomon people, we analyzed coprolites (7000–5500 years ago) excavated from the Torihama shell-mound site, located in the middle of the Japanese archipelago. Significant numbers of reads obtained by shotgun sequencing were aligned to known microbial genomes reported in the modern human gut environment. We detected viruses (mostly phages) and their host-derived sequences simultaneously, in order to analyze the host–viral relationships that seem to have been maintained for more than thousands of years. We also obtained genomic sequences from what appears to be ingested foods, such as salmon.

## Materials and methods

### Sampling

More than 400 coprolites were excavated from the Torihama shell-mound site in Fukui Prefecture, Japan, during the 1970s [15]. We sampled ten of these, which were estimated to belong the Early Jomon period (7000–5500 BP) according to the stratigraphy. We used four of the ten coprolites, which showed relatively high-concentration NGS libraries, for metagenome shotgun sequencing analyses (S1 Table). All necessary permits were obtained for the described study, which complied with all relevant regulations. We obtained research material usage approval from the Wakasa History Museum in Fukui Prefecture, Japan, on July 26, 2018.

### DNA extraction and library preparation

DNA extraction and library preparation were carried out in a clean-room facility dedicated to ancient DNA research at Kitasato University and University of Tokyo. Pieces weighing 2–5 g were extracted from each coprolite sample using a sterilized surgical knife. The outer surface of the coprolite sample was removed (with 1–2 mm thickness), after which UV radiation of the sample was performed for 30 minutes, in order to reduce the exogenous DNA contamination risk. Coprolite samples were frozen with liquid nitrogen and crushed into a fine powder using a ShakeMaster Auto ver.2.0 (BioMedical Science), and DNA extraction was conducted using NucleoSpin DNA Stool (MACHEREY-NAGEL) with 0.08–0.1 g of powdered sample. Four DNA extracts from TH55, TH58, TH62, and TH74 containing 5.80 ng, 4.46 ng, 15.54 ng, and 5.36 ng of DNA, respectively, were proceeded to library preparation for metagenomic sequencing. Library construction for the Illumina sequencer was conducted using an NEBNext Ultra II DNA Library Prep Kit for Illumina and NEBNext Multiplex Oligos for Illumina (96 Unique Dual Index Primer Pairs; New England Biolabs) with 1 ng of DNA template. After size selection for excluding larger DNA fractions containing possible contaminant DNAs with x0.9 volume of AMPure XP Beads (Beckman Coulter Genomics, USA), library amplification was

performed, with 10 to 12 cycles. Sequencing was performed on the Illumina NovaSeq6000 platform using a 150 bp paired-end dual-index run with 300 cycles.

## Data pre-processing

First, we removed adapters and phiX174 spiked in reads using BBDuk with the following options: ftm = 5; qtrim = rl; trimq = 20; ftl = 15; ref = adapters,phix; ktrim = r; k = 23; mink = 11; hdist = 1; tpe tbo maq = 20. The human-derived reads were aligned to the hg19 human reference genome using the Burrows-Wheeler Aligner (BWA) 0.7.17 [16] and discarded. The human-decontaminated reads were then deduplicated using Dedupe. Finally, the remained reads were error-corrected with Tadpole using the mode = correct and k = 50 options. BBDuk, Dedupe, and Tadpole were provided by BBTools 38.49 (BBMap; Bushnell, B.; sourceforge.net/projects/bbmap/).

## Viral search

We collected viral sequences, including bacteriophages, from the NCBI viral RefSeq (released September 2020) [17, 18] and the Millard labs PHAge REference Database (INPHARED; July 2019) [19]. We also utilized gut viral sequences from the human Gut Virome Database (GVD) [20] and an in-house data set of CRISPR targeted viral sequences constructed from human gut metagenomic data [21]. All pre-processed reads were aligned to these viral reference genomes using the BWA. After alignment, reads with mapping quality below 25 and duplicated ones were filtered out using SAMtools 1.10 [22]. The read length distribution and coverages were analyzed using BEDtools 2.29.2 [23]. We then calculated the post-mortem degradation score (PMDS) for each filtered read using PMDtools, in order to distinguish the ancient DNA from modern contaminants [24]. The viral classification was analyzed based on the taxonomic classification of the GVD. We used the old taxonomic classification information in the GVD, although several major families such as *Siphoviridae*, *Myoviridae*, and *Podoviridae* were removed from the International Committee on Taxonomy of Viruses (ICTV) classification in August 2022 [25].

## Detection of viral host genomes and food remnants

We utilized the genomes of 19 host bacterial species of the viruses detected in the previous study [21]. Genomic assembly data of *Acanthamoeba castellanii* (GenBank assembly accession was GCA_000313135.1) was downloaded from the NCBI database. We also downloaded 4644 human gut prokaryotic genomes and their taxonomic information from the Unified Human Gastrointestinal Genome (UHGG) collection [26]. A maximum-likelihood phylogenetic tree of the 4616 bacterial species detected in the human gut was also downloaded from UHGG collection. The phylogenetic tree was visualized by iTOL v6 and added the gray bar graphs in the outermost layer, representing the reads ratio aligned to bacterial genomes in all four samples [27]. To analyze food remnants, we searched candidate sequences based on the BLAST search using NCBI non-redundant nucleotide database (downloaded on 23 February 2021) [17, 28]. According to the BLAST results, we downloaded genome assembly data of *Oncorhynchus nerka* (RefSeq assembly accession was GCF_006149115) and *Vigna angularis* (RefSeq assembly accession was GCF_001190045). All pre-processed reads were mapped to those reference sequences and filtered as previously described.

## Results and discussion

We extracted DNA from four coprolites found at an archaeological site from the Early Jomon period and conducted shotgun metagenomic sequences using the Illumina NovaSeq6000. We

**Table 1. Number of detected gut viral and bacterial genomes.**

| Reference name | Number of references | Detected references | Detected references (PMDS filtered) |
|---|---|---|---|
| GVD (gut virome database) | 33,242 | 32,989 | 17,919 |
| CRISPR targeted phages | 11,391 | 5,236 | 1,410 |
| Host bacteria of CRISPR targeted phages | 19 | 19 | 19 |
| Unified Human Gastrointestinal Genome (UHGG) | 4,644 | 4,644 | 4,613 |

obtained 4,944,613,662 reads in total (S2 Table). After pre-processing (including human decontamination), 3,188,801,202 reads remained, which were utilized for downstream analyses. During human decontamination, the human genome (hg19) alignment results revealed that the average and median alignment lengths were extremely short—both around 20 bp—reflecting serious fragmentation of host DNAs in the fecal sample over time (S3 Table). About 1% of the deduplicated reads showed post-mortem signatures, based on a postmortem degradation score (PMDS) greater than 3 [24].

We first analyzed the viral genomes. Viral genomes are typically shorter than those of other organisms; therefore, it was considered more efficient to detect their genomes with highly fragmented reads. We aligned the reads to viral genomes obtained from the modern human gut environment, in order to focus on the viruses existing in the gut and avoid soil contamination. We utilized 33,242 unique viral sequences and taxonomic information from the Gut Virome Database (GVD) [20], detecting 17,919 GVD-derived sequences with postmortem degradation (Table 1). Based on the taxonomic classification of the GVD, the results demonstrated that most of the viral sequences originated from bacteriophages (Fig 1A). Within the bacteriophages, the *Siphoviridae*, *Myoviridae*, and *Podoviridae* families were abundant (Fig 1B), which are also observed in the gut virome of present-day humans [20, 29, 30]. In the GVD alignment results, we removed the sequence reads that were filtered out for PMDS and showed larger than 30 bp average and median alignment length, as these reads might be derived from contamination, based on the average and median read length in human genome alignment. The maximum coverage was 77%, and 31 viral species genomes showed more than 50% coverage (S4 Table). The average alignment length was about 20 bp, reflecting the highly fragmented DNA, which might be related to the state of preservation and gut environment. Nevertheless, a vast number of reads from the Jomon coprolites were mapped to gut viral sequences. Although sequence variations were not detectable, due to the low depth of alignments, there might be

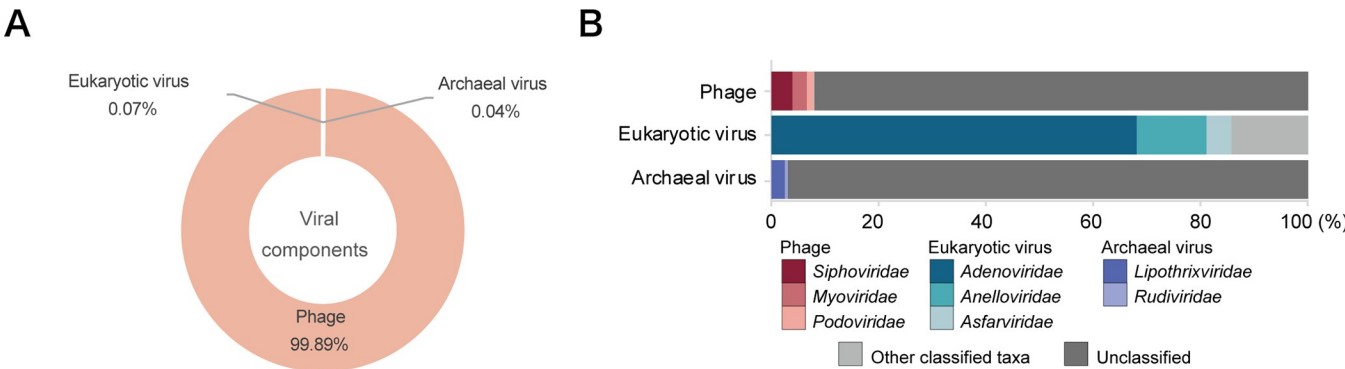

**Fig 1. Viral components of Jomon people's coprolites.** (A) The pie chart indicates the ratio of archaeal viruses, eukaryotic viruses, and phages based on reads alignment results. The taxonomic information of GVD was utilized for classification. (B) Bar charts show the taxonomic composition of each viral group based on an abundance of each viral family-derived reads. The colored families indicate the top three abundant ones. The light gray bar contains reads of all other known viral families, and the dark gray bar contains reads of unclassified viruses.

**Table 2. Alignment to eukaryotic viral genomes.**

| Sample | Reference | Reference length (bp) | Average reads length (bp) | Median reads length (bp) | Number of reads (deduplicated) | Number of reads (PMDS≥3) | Coverage% (1<x)* |
|---|---|---|---|---|---|---|---|
| TH55 | Human herpesvirus 5 (NC_006273.2) | 235,646 | 19.74 | 19 | 6,201 | 64 | 15.61 |
| TH58 | Human herpesvirus 5 (NC_006273.2) | 235,646 | 19.80 | 19 | 5,294 | 62 | 13.87 |
| TH62 | Human herpesvirus 5 (NC_006273.2) | 235,646 | 19.77 | 19 | 4,528 | 40 | 12.24 |
| TH74 | Human herpesvirus 5 (NC_006273.2) | 235,646 | 19.74 | 19 | 6,682 | 67 | 16.67 |
| TH55 | Human adenovirus F (NC_001454.1) | 34,214 | 19.73 | 19 | 261 | 4 | 5.76 |
| TH58 | Human adenovirus F (NC_001454.1) | 34,214 | 19.67 | 19 | 228 | 5 | 5.00 |
| TH62 | Human adenovirus F (NC_001454.1) | 34,214 | 19.69 | 19 | 177 | 5 | 4.09 |
| TH74 | Human adenovirus F (NC_001454.1) | 34,214 | 19.66 | 19 | 266 | 2 | 6.31 |
| TH55 | Pandoravirus dulcis (NC_021858.1) | 1,908,524 | 19.85 | 19 | 117,362 | 1,296 | 29.80 |
| TH58 | Pandoravirus dulcis (NC_021858.1) | 1,908,524 | 19.87 | 19 | 98,447 | 1,261 | 26.43 |
| TH62 | Pandoravirus dulcis (NC_021858.1) | 1,908,524 | 19.85 | 19 | 85,782 | 1,057 | 24.11 |
| TH74 | Pandoravirus dulcis (NC_021858.1) | 1,908,524 | 19.85 | 19 | 131,411 | 1,380 | 32.01 |
| TH55 | Mollivirus sibericum isolate P1084-T (NC_027867.1) | 651,523 | 19.75 | 19 | 20,697 | 210 | 19.30 |
| TH58 | Mollivirus sibericum isolate P1084-T (NC_027867.1) | 651,523 | 19.78 | 19 | 16,699 | 178 | 16.45 |
| TH62 | Mollivirus sibericum isolate P1084-T (NC_027867.1) | 651,523 | 19.76 | 19 | 14,757 | 148 | 15.03 |
| TH74 | Mollivirus sibericum isolate P1084-T (NC_027867.1) | 651,523 | 19.74 | 19 | 22,137 | 197 | 20.51 |

*Using deduplicated reads

similar or closely related viral species with the present-day gut viruses as those in the gut of Jomon people. To confirm this hypothesis, the additional sequencing of well preserved coprolites DNA might be required to obtain the species-level viral genomic sequences and to compare the modern viral genomes. Based on the previous reports, the gut viral composition differs across the life stages of human [20]. For example, eukaryotic viruses are more prevalent in infancy, due to the undeveloped immune system, but remain low throughout the rest of life. Our data illustrated a high abundance of bacteriophages, especially *Siphoviridae* viruses. Therefore, the coprolites might reflect the ancient gut environment of the Jomon people.

As the detected viral classification indicated the existence of eukaryotic viral genomes in the Jomon coprolites, we examined the eukaryotic viral genomes registered in the NCBI database. We observed reads that aligned to several human viral genomes and giant viral genomes in the coprolite data as shown in Table 2. Meanwhile, we detected the reads showing homology with human pathogenic viruses, such as human betaherpesvirus 5 and human adenovirus F, in all the samples indicating the existence of these viruses in the bodies of the Jomon people.

**Table 3. Alignment to amoeba genome.**

| Sample | Average reads length (bp) | Median reads length (bp) | Number of reads | Number of reads (PMDS≥3) | Coverage% (1<x)* |
|--------|---------------------------|--------------------------|-----------------|--------------------------|------------------|
| TH55 | 20 | 19 | 1,545,185 | 15,045 | 18.5061 |
| TH58 | 20 | 19 | 1,273,715 | 15,134 | 16.1005 |
| TH62 | 20 | 19 | 1,111,251 | 12,205 | 14.7077 |
| TH74 | 19.9 | 19 | 1,651,614 | 15,168 | 19.3704 |

*Using deduplicated reads

Two different giant viral genomes—Pandoravirus and Mollivirus—infecting *Acanthamoeba castellanii* were detected (Table 2). Giant viruses have been discovered since 2003 [31], which have large genomes (>100 kb) and more than 500 protein-coding sequences [32–34]. The hygiene environments of the Jomon are not yet well known, but the detection of the giant viral genome in the Jomon coprolite may possibly suggest that they had poorer hygiene than modern humans. We also attempted to detect Acanthamoeba genomes, which are the hosts of the giant viruses, and more than one million reads were aligned to the amoeba genome with more than 10% coverage, indicating the existence of amoeba genomes in the Jomon coprolites as shown in Table 3.

These results indicated the possibility of simultaneously detecting host and viral genomes in the coprolite samples. We attempted to detect both gut phages and their bacterial host genomes, in order to examine their cohabitation. Based on our previous works, we detected 11,391 terminally redundant sequences targeted by host clustered regularly interspaced short palindromic repeats (CRISPR) immunological memory [21]. As prokaryotic cells can memorize previously infected phage sequences in their CRISPR system, we were able to determine the candidate hosts, based on the connection between the CRISPR-targeted sequences, or protospacers, and the associated CRISPR direct repeats. As a result, we detected 1410 CRISPR-targeted phages (Table 1), and the overall coverage ranged up to 40% (S5 Table). Within the 1410 detected phages, 23 candidate hosts were estimated, based on our previous results [21]. Then, we mapped all reads to 19 available reference sequences, resulting in the detection of 19 bacterial sequences (S6 Table). Thus, we confirmed both phage and host bacterial genome sequences in the coprolites, analogous to the giant virus and amoeba cases. Our results demonstrated that the host and viral co-habitants in the Jomon gut environment might be conserved through their evolutionary history.

As we could align several reads to gut bacterial genomes, we then decided to utilize all representative gut bacterial genomes for alignment analyses. We utilized the 4644 gut prokaryotes from the Unified Human Gastrointestinal Genome (UHGG) collection [26]. Some gut bacteria, such as Bradyrhizobium bacteria, also live in soil environments, and may have contaminated the coprolites over time. In the present analyses, sequence reads aligned to soil bacteria had a tendency to show longer lengths due to lower fragmentation; thus, those with average alignment length longer than 30 bp—which were less likely from the Jomon DNAs—were filtered out. As a result, 4613 gut bacterial genomes were detected, as shown in Table 1 and S7 Table. We analyzed the bacterial compositions and the phylogenetic relationship of the detected gut bacterial species using the phylogenetic tree reported in [26] (Fig 2 and S1 Fig). In Fig 2, the bar graphs indicate the abundance of reads of bacterial genome-aligned reads within the four Jomon coprolite samples. The read abundance of each species was less than 0.5%. On the other hand, several species of Proteobacteria and Actinobacteria bacteria showed higher than 0.2% abundance. These bacterial phyla were highly prevalent, as well as Firmicutes A, Bacteroidota, and Verrucomicrobiota bacteria (S1 Fig). These tendencies could be seen in the present-day human gut microbiome [35].

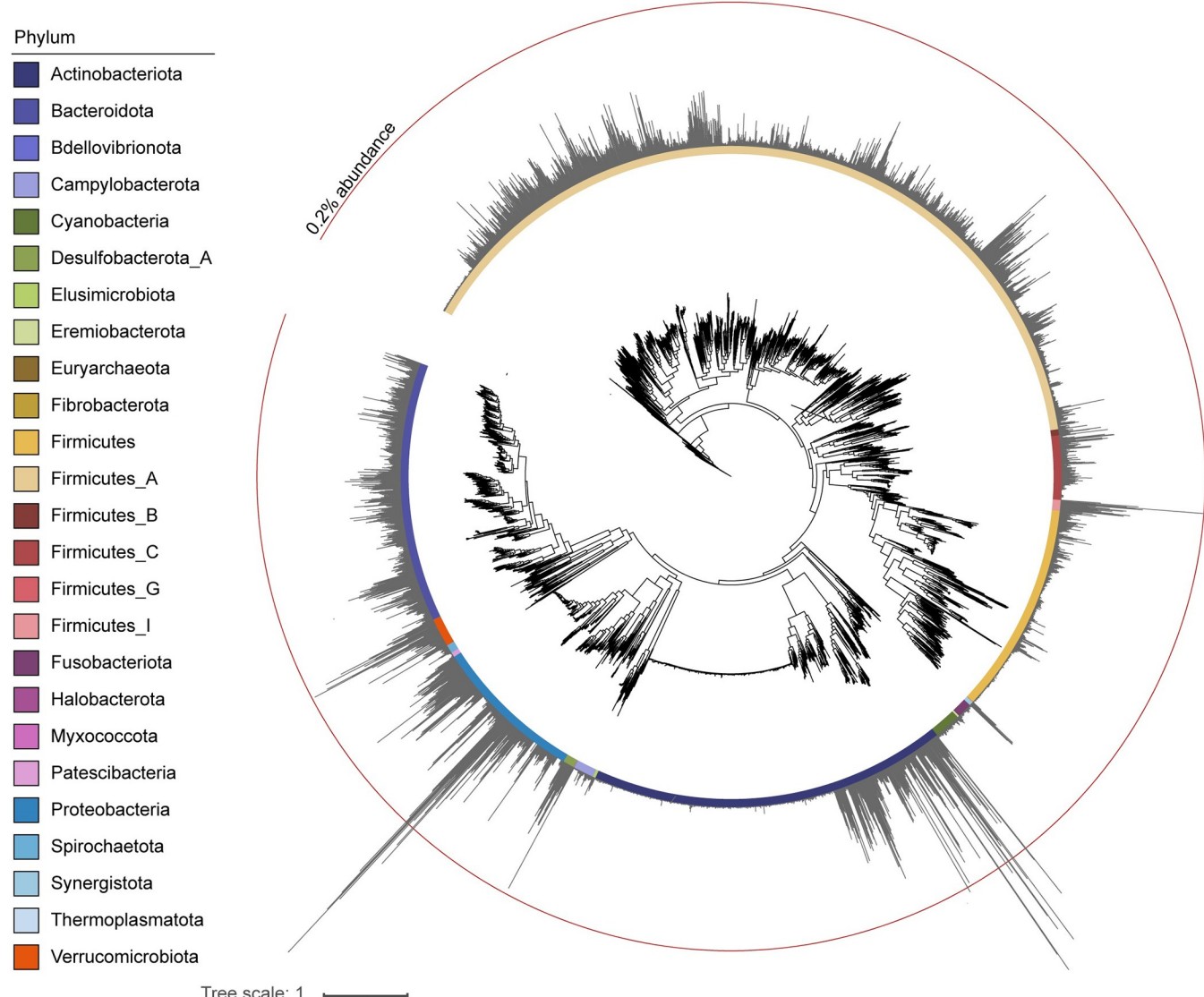

**Fig 2. Phylogenetic trees of detected gut bacteria.** The tree indicates phylogenetic relationships of 4,616 bacterial species. The colored chart shows different bacterial phylum. The phylogenetic tree and each taxonomic information were derived from the previous report [26]. The gray bar graphs in the outermost layer represent the reads ratio of reads aligned to bacterial genomes in all four samples. The red circular line is a scale bar indicating 0.2% abundance of aligned reads.

Finally, we analyzed genomic information from the candidate foods, in order to infer the cultural and behavioral characteristics of the Torihama Jomon people. We detected the alignment of reads to the *Oncorhynchus nerka* (salmon) and *Vigna angularis* (red beans) genomes (S8 and S9 Tables). A previous archeological report has indicated that the Torihama Jomon people hunted freshwater and saltwater fish, such as salmon, based on the lipids detected from Jomon pottery [36]. Our results provided genomic support that the Jomon people may have hunted and used salmon (probably, red salmon) as a food resource, at least with respect to the Torihama shell-mound site. Further detailed analyses would be required to find robust evidence that it is a salmon.

In conclusion, we obtained genomic data from the coprolites of Jomon people and conducted alignment of reads to the genomes of gut viruses, gut microbes, and food remnants.

Even though the average length of the reads was too short to analyze genomic variations, we still observed a vast number of reads mapped to the genomes of viruses, bacteria, and likely foods. Based on our alignment results, we could estimate the existence of microbes and host–viral coexistence in the Jomon gut environment, comparable to that in present-day humans, suggesting that the coprolites have the potential to reveal long-term host–viral co-evolution trends. Furthermore, we discovered the reads of possible foods through genomic information, providing biological evidence for the dietary characteristics of the Jomon people.

## Supporting information

**S1 Fig. Bacterial compositions of Jomon people's coprolites.** Bar charts show the taxonomic classification of detected gut bacteria and proportion of each bacterial groups. The ratio was calculated based on the number of aligned reads. The colored bars indicate the top five abundant classifications. The light gray bar contains reads of all other known bacterial families, and the dark gray bar contains reads of unclassified bacteria.
(TIF)

**S1 Table. Sample information.**
(XLSX)

**S2 Table. Preprocess results.**
(XLSX)

**S3 Table. Human genome alignment.**
(XLSX)

**S4 Table. Alignment to GVD sequences.**
(XLSX)

**S5 Table. Alignment to CRISPR targeted sequences.**
(XLSX)

**S6 Table. Alignment to host bacterial genomes.**
(XLSX)

**S7 Table. Alignment to gut bacterial genomes.**
(XLSX)

**S8 Table. Alignment to a salmon genome.**
(XLSX)

**S9 Table. Alignment to a red bean genome.**
(XLSX)

## Acknowledgments

We thank all the members of the Human Genetics Laboratory at the National Institute of Genetics for providing many suggestions. Most of the computational analyses were performed on the NIG supercomputer at ROIS National Institute of Genetics. This work was supported in part by The Graduate University for Advanced Studies, SOKENDAI. We would like to thank MDPI's language editing service for English editing.

## Author Contributions

**Conceptualization:** Luca Nishimura, Ituro Inoue.

**Data curation:** Akio Tanino.

**Formal analysis:** Luca Nishimura.

**Funding acquisition:** Luca Nishimura, Ryota Sugimoto, Hiroki Oota, Ituro Inoue.

**Investigation:** Luca Nishimura, Akio Tanino, Hirofumi Nakaoka.

**Methodology:** Luca Nishimura, Akio Tanino, Daisuke Waku, Masahiko Kumagai, Ryota Sugimoto.

**Project administration:** Hiroki Oota, Ituro Inoue.

**Resources:** Mayumi Ajimoto.

**Software:** Luca Nishimura.

**Supervision:** Takafumi Katsumura, Motoyuki Ogawa, Kae Koganebuchi, Daisuke Waku, Masahiko Kumagai, Hiroki Oota, Ituro Inoue.

**Validation:** Ryota Sugimoto, Hiroki Oota.

**Visualization:** Luca Nishimura.

**Writing – original draft:** Luca Nishimura.

**Writing – review & editing:** Masahiko Kumagai, Ryota Sugimoto, Hirofumi Nakaoka, Hiroki Oota, Ituro Inoue.

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
