## [Decision Letter · Decision Letter 0]

12 May 2023

PONE-D-23-06645Metagenomic analyses of 7000 to 5500 years old coprolites excavated from the Torihama shell-mound site in the Japanese archipelagoPLOS ONE

Dear Dr. Inoue,

Thank you for submitting your manuscript to PLOS ONE. After careful consideration, we feel that it has merit but does not fully meet PLOS ONE’s publication criteria as it currently stands. Therefore, we invite you to submit a revised version of the manuscript that addresses the points raised during the review process. Specifically, considering the damaged DNA, short read mapping rather than long contigs, both reviewers have raised the concerns about the "presence" of the the viruses in the ancient samples. This should be resolved. Besides, more details of the methods should be provided.

We look forward to receiving your revised manuscript.

Kind regards,

Yanpeng Li, Ph.D.

Academic Editor

PLOS ONE

2. In your manuscript, please provide additional information regarding the specimens used in your study. Ensure that you have reported specimen numbers and complete repository information, including museum name and geographic location.

For more information on PLOS ONE's requirements for paleontology and archaeology research, see https://journals.plos.org/plosone/s/submission-guidelines#loc-paleontology-and-archaeology-research.

Reviewers' comments:

Reviewer's Responses to Questions

**Comments to the Author**

1. Is the manuscript technically sound, and do the data support the conclusions?

Reviewer #1: Yes

Reviewer #2: Partly

2. Has the statistical analysis been performed appropriately and rigorously? 

Reviewer #1: N/A

Reviewer #2: N/A

3. Have the authors made all data underlying the findings in their manuscript fully available?

Reviewer #1: Yes

Reviewer #2: No

4. Is the manuscript presented in an intelligible fashion and written in standard English?

Reviewer #1: Yes

Reviewer #2: Yes

5. Review Comments to the Author

Reviewer #1: In this study, the authors analyzed coprolites (7000–5500 years ago) excavated from the Torihama shell-mound site and found sequences closely related to known gut microbe, viruses, and food genomes, prompting a better understanding of the gut environment and lifestyle of ancient peoples. Though this paper is interesting, I found there were missing details of method, unsound conclusion and many other drawbacks.

Major comments

1. Line 63: “genome information from such remains could be available if the endogenous DNAs are in a good state of preservation”. So, how do the authors determine the preservation state of the coprolites in the present study?

2. Line 89: The authors sampled ten coprolites from 400 of them, and then they selected four of the ten coprolites showed relatively high concentration. Could the authors elaborate on the sampling process and the specific selection standard?

3. Line 168: The authors aligned the reads to viral genomes obtained from the modern human gut environment to focus on the viruses existing in the gut and avoid soil contamination. The microorganisms inhabiting in gut and soil are not totally different, so doing alignment with viral genomes obtained from the modern human gut can’t completely remove the contaminants from soil. More measures should be taken to deal with this issue.

4. Line 181: “The average alignment length was about 20 bp, reflecting the highly fragmented DNA”. As the alignment length was too short, how did the authors ensure accuracy during the alignment process?

5. Line 185: “there might be similar or closely related viral species with the present-day gut viruses as those in the gut of Jomon people.” The fact that the research outcomes in the present study were not enough to infer the existence of similar or closely related viral species in the coprolites. Authors should provide more evidence to support this statement.

6. Line 206: “we detected human pathogenic viruses, such as human betaherpesvirus 5 and human adenovirus F”. As the coverages of the viral alignment results were extremely low (around 10%), how did the author get the species classification information of the viral sequences detected in the study? Please provide more information about the classification standard.

7. Line 256: “This pattern seems to correspond to the present-day human gut microbiome”. There was no comparison analysis of the gut microbiome between ancient and present-day human in the study, so the authors should provide more evidence and references to support this conclusion.

Minor comments

1. Line 52: “virus” should be “viruses”

2. Line 83: “more than several thousands of years” should be “more than thousands of years”

3. Line 89: “We sampled ten of these, estimated to belong the Early Jomon period” should be “We sampled ten of these, which were estimated belong to the Early Jomon period”

4. Line 120: “error corrected” should be “error-corrected”

5. Line 174: The viral family names should be written in italics.

6. The tables in the study should follow three-line table format.

7. The figure legends should be placed at the end of the manuscript together.

8. Line 213: “Two different giant viral genomes—Pandoravirus and Mollivirus—infecting Acanthamoeba castellanii were detected (Table 3)”. Table 3 presents the results about amoeba and doesn’t contain any information about the two giant viruses. Please provide the corresponding results here.

9. Line 220: “aligned to cover more than 10% of the amoeba genome” should be “aligned to the amoeba genome with more than 10% coverage”

10. Line 226: “host bacterial genomes” should be “bacterial host genomes”

11. Line 261: “The phylogenetic tree and each taxonomic information were derived from the previous report”. This sentence means that the phylogenetic tree in fig. 2 was generated from pervious study not the present study. Please confirm the statement regarding the result.

12. Line 274: “that it is a salmon” should be deleted.

Reviewer #2: This study used short-read metagenomes to evaluate the possible presence of ancient organisms and food in the coprolites. Overall, this work is interesting, particularly due to those four valuable metagenomic datasets and the aim to shedding light on human dietary habits in ancient times.

However, the authors should establish a more rigorous threshold (e.g., a convinced read identity, read coverage, and genome coverage in mapping) for evaluating the "presence or absence" of a seed genome/organism in the coprolite metagenomes. In addition, the study is limited by concerns related to ancient DNA damage and sequencing errors, which can affect read mapping and biological inferences. Furthermore, the paper should benefit greatly from additional analyses, such as assembly, binning, and direct identification of ancient viral and microbial genomes from the four metagenomes.

6. PLOS authors have the option to publish the peer review history of their article (what does this mean?). If published, this will include your full peer review and any attached files.

Reviewer #1: No

Reviewer #2: **Yes: **Zhi-Ping Zhong

---

## [Author Response · Author response to Decision Letter 0]

9 Jun 2023

Response to Reviewers 

To Reviewer #1 

In this study, the authors analyzed coprolites (7000–5500 years ago) excavated from the Torihama shell-mound site and found sequences closely related to known gut microbe, viruses, and food genomes, prompting a better understanding of the gut environment and lifestyle of ancient peoples. Though this paper is interesting, I found there were missing details of method, unsound conclusion and many other drawbacks.

Reply:

We thank the reviewer for several critical and valuable comments, which would much improve the manuscript. We revised the manuscript according to the reviewer’s comments. 

Major comments

1. Line 63: “genome information from such remains could be available if the endogenous DNAs are in a good state of preservation”. So, how do the authors determine the preservation state of the coprolites in the present study?

Reply:

The coprolites used in this study were excavated by the Torihama shell-mound site in the 1970s and then stored in a dark storage room at the Wakasa History Museum. We sampled from this storage room. However, what we describe in the text as being “in a good state of preservation” is not after excavation. These coprolites were found in sediments with alternating peat and shell layers. We believe that this relatively alkaline and oxygen-free environment contributed to the better preservation of DNA in coprolites.

2. Line 89: The authors sampled ten coprolites from 400 of them, and then they selected four of the ten coprolites showed relatively high concentration. Could the authors elaborate on the sampling process and the specific selection standard?

Reply:

Of the 400 coprolites, the best looking ones were not analyzed for display in the museum. The 10 worst-looking ones were randomly selected. DNA was extracted from these and the concentration was measured. We performed PCR using these as templates and performed amplicon sequencing (to be published elsewhere). After the PCR amplicon sequencing, we had four individuals that still DNA left. In this study, libraries were constructed using these four individuals with relatively high DNA concentrations and used for shotgun sequencing.

3. Line 168: The authors aligned the reads to viral genomes obtained from the modern human gut environment to focus on the viruses existing in the gut and avoid soil contamination. The microorganisms inhabiting in gut and soil are not totally different, so doing alignment with viral genomes obtained from the modern human gut can’t completely remove the contaminants from soil. More measures should be taken to deal with this issue.

Reply:

We set several criteria to reduce the number of contaminated reads based on the result of human genome alignment. We removed reads with low mapping quality, duplication, more than 30 bp alignment length, since the average alignment length of human reads corresponding to Jomon individual was about 20 bp. Evidently, heavily fragmented reads are not preferable for mapping, but we can consider that the longer reads are derived from post-Jomon period. We also removed reads with low postmortem degradation score (PMDS), which might be from contaminations. We set a threshold of PMDS ≥ 3 based on the previous report, Skoglund et al. 2014 (ref 24). We believe that our protocol is not the ideal one but a practical method to reduce the modern contaminated sequences. 

Comparison between the genomic sequencing data of coprolites and the soils surrounding coprolites might elucidate the contamination from the soil as a future work.

4. Line 181: “The average alignment length was about 20 bp, reflecting the highly fragmented DNA”. As the alignment length was too short, how did the authors ensure accuracy during the alignment process?

Reply:

After the alignment to reference genomes, reads with mapping quality below 25 and duplicated ones were filtered out as mentioned in the methods section. We also constructed the scrambled sequences of reference genomic sequences with the same nucleotide composition and the same length by Python random module to see the alignment randomness. We then aligned all the reads against the scrambled sequences using BWA. For example, the number of aligned reads differed ten times between human or viral reference genomes and scrambled sequences in each coprolite sample. For an instance, in the TH55 sample, the number of aligned reads to the human genome was about 219 millions, and to a scrambled sequence was about 29 millions before filtering. Accordingly, the uniqueness of mapped reads was supported by the alignment results. On the other hand, when we conducted the same analyses with Bowtie2, we could not find any differences between reference genome alignments and scrambled ones. 

5. Line 185: “there might be similar or closely related viral species with the present-day gut viruses as those in the gut of Jomon people.” The fact that the research outcomes in the present study were not enough to infer the existence of similar or closely related viral species in the coprolites. Authors should provide more evidence to support this statement.

Reply:

We thank the reviewer for raising a critical point that the current results were insufficient to conclude the presence of similar or closely related viral species in the coprolites. It is indeed hard to provide more evidence to support it with additional analyses of our current sequencing data. Future studies with highly preserved coprolite DNA might help to obtain ancient viral sequences at species-level resolution. We added the following sentence: To confirm this hypothesis, the additional sequencing of highly preserved coprolites DNA might be required to obtain the species-level viral genomic sequences and to compare the modern viral genomes.

6. Line 206: “we detected human pathogenic viruses, such as human betaherpesvirus 5 and human adenovirus F”. As the coverages of the viral alignment results were extremely low (around 10%), how did the author get the species classification information of the viral sequences detected in the study? Please provide more information about the classification standard.

Reply:

We conducted the reads alignment to the human viral reference genomes, and we observed that several reads were aligned to the reference genomes. Since our description might cause misleading, we corrected sentences in the same paragraph as follows: As the detected viral classification indicated the existence of eukaryotic viral genomes in the Jomon coprolites, we examined the eukaryotic viral genomes registered in the NCBI database. We observed reads that aligned to several human viral genomes and giant viral genomes in the coprolite data as shown in Table 2. Meanwhile, we detected the reads showing homology with human pathogenic viruses, such as human betaherpesvirus 5 and human adenovirus F, in all the samples indicating the existence of these viruses in the bodies of the Jomon people.

7. Line 256: “This pattern seems to correspond to the present-day human gut microbiome”. There was no comparison analysis of the gut microbiome between ancient and present-day human in the study, so the authors should provide more evidence and references to support this conclusion.

Reply:

We thank the reviewer pointing out the overstatement of our results. Based on our read abundance data, bacterial phyla Firmicutes A, Actinobacteriota, Proteobacteria, Bacteroidota, and Verrucomicrobiota bacteria were highly prevalent. These tendencies were seen in previous reports such as Arumugam et al. 2011. We modified the sentence and added the reference at the end of the sentence. 

Minor comments

1. Line 52: “virus” should be “viruses”

Reply:

We corrected the word from “virus” to “viruses”.

2. Line 83: “more than several thousands of years” should be “more than thousands of years”

Reply:

We corrected the sentence from “more than several thousands of years” to “more than thousands of years”.

3. Line 89: “We sampled ten of these, estimated to belong the Early Jomon period” should be “We sampled ten of these, which were estimated belong to the Early Jomon period”

Reply:

We corrected the sentence from “We sampled ten of these, estimated to belong the Early Jomon period” should be “We sampled ten of these, which were estimated belong to the Early Jomon period”. 

4. Line 120: “error corrected” should be “error-corrected”

Reply:

We corrected the word from “error corrected” should be “error-corrected”.

5. Line 174: The viral family names should be written in italics.

Reply:

We wrote viral family names in italics. We also corrected the style of viral family names at Line 137, 190 and Fig 1B. 

6. The tables in the study should follow three-line table format.

Reply:

We rechecked the formatting rules of PLOS ONE to make tables (https://journals.plos.org/plosone/s/tables). However, we could not find a requirement of following three-line table format. 

7. The figure legends should be placed at the end of the manuscript together.

Reply:

We rechecked the PLOS ONE submission guidelines (https://journals.plos.org/plosone/s/submission-guidelines) and confirmed the requirement as below: “Figure captions must be inserted in the text of the manuscript, immediately following the paragraph in which the figure is first cited (read order)”.

8. Line 213: “Two different giant viral genomes—Pandoravirus and Mollivirus—infecting Acanthamoeba castellanii were detected (Table 3)”. Table 3 presents the results about amoeba and doesn’t contain any information about the two giant viruses. Please provide the corresponding results here.

Reply:

We miswrote the Table number. We corrected the number from Table 3 to Table 2. 

9. Line 220: “aligned to cover more than 10% of the amoeba genome” should be “aligned to the amoeba genome with more than 10% coverage”

Reply:

We corrected the sentence from “aligned to cover more than 10% of the amoeba genome” to “aligned to the amoeba genome with more than 10% coverage”.

10. Line 226: “host bacterial genomes” should be “bacterial host genomes”

Reply:

We corrected sentence from “host bacterial genomes” to “bacterial host genomes”. 

11. Line 261: “The phylogenetic tree and each taxonomic information were derived from the previous report”. This sentence means that the phylogenetic tree in fig. 2 was generated from pervious study not the present study. Please confirm the statement regarding the result.

Reply:

We utilized the phylogenetic information of 4,616 bacterial genomes derived from Almeida et al. 2021 and visualize the phylogenetic tree by iTOL v6. We added the gray bar graphs in the outermost layer which represent the reads ratio of reads aligned to bacterial genomes in all four samples. We added the above description at lines 148-150. 

12. Line 274: “that it is a salmon” should be deleted.

Reply:

We deleted the sentence.  

To Reviewer #2

This study used short-read metagenomes to evaluate the possible presence of ancient organisms and food in the coprolites. Overall, this work is interesting, particularly due to those four valuable metagenomic datasets and the aim to shedding light on human dietary habits in ancient times.

However, the authors should establish a more rigorous threshold (e.g., a convinced read identity, read coverage, and genome coverage in mapping) for evaluating the "presence or absence" of a seed genome/organism in the coprolite metagenomes. In addition, the study is limited by concerns related to ancient DNA damage and sequencing errors, which can affect read mapping and biological inferences. Furthermore, the paper should benefit greatly from additional analyses, such as assembly, binning, and direct identification of ancient viral and microbial genomes from the four metagenomes.

Reply:

First of all, we removed the sequence reads by filtering out for postmortem damage score (PMDS) ≥ 3 and showing larger than 30 bp average and median alignment length, as these reads were likely derived from contamination based on the average and median read length in human genome alignment. Because of the short alignment length, we utilized PMDtools to distinguish ancient DNA from modern contaminants. Skoglund et al. 2014 (ref 24) imposed a tight PMDS threshold (PMDS≥5) to decrease the contaminated sequences, but the tight threshold caused the removal of the ancient sequences. So, we chose PMDS ≥ 3 as our threshold to remove contaminations and remain ancient sequences. 

Furthermore, we constructed scrambled sequences of reference genomic sequences with the same nucleotide composition and the same length by Python random module to see the alignment randomness. We then aligned all the reads against the scrambled sequences using BWA. For example, the number of aligned reads differed ten times between human or viral reference genomes and scrambled sequences in each coprolite sample. For instance, in the TH55 sample, the number of aligned reads to the human genome was about 219 millions, and to a scrambled sequence was about 29 millions before filtering. The alignment results confirmed the uniqueness of mapped reads. Comparable patterns were noted when we employed randomized sequences of RefSeq viral genomes and gut viral references in conjunction with each coprolite sample. On the other hand, when we conducted the same analyses with Bowtie2 (with following options: --end-to-end --very-sensitive -N 1), we could not find any differences between reference genome alignments and scrambled ones. The dissimilar outcomes could be attributed to the contrasting alignment algorithms employed and options. For instance, BWA utilizes the Ferragina and Manzini matching algorithm to locate precise matches, whereas Bowtie utilizes a modified version of the Ferragina and Manzini matching algorithm to determine the mapping location (Hatem et al., 2013). 

We tested the viability of assembly with the four coprolites samples by collecting the human-aligned reads and conducted de novo assembly with SPAdes. However, we could not reconstruct the human genomes and detected several misassembled contigs. It might be caused by DNA fragmentation. We then suspected that it seems to be difficult to reconstruct ancient viral genomes by assembly because viral sequences were also fragmented like human genomes, therefore we did not conduct de novo assembly to obtain ancient viral genomes. In addition, we did not test binning because of the fragmented sequences. Several binning methods such as metabat (Kang et al. 2019), require ≥1,500 bp sequences as input and need de novo assembly in advance. Other binning methods use k-mers reflecting nucleotide compositions directly using short reads (Kyrgyzov et al. 2020). Raw reads are parsed into k-mers of fixed size like k=31, requiring longer sequences. 

We understand that “direct identification” means methods such as target capture sequencing and PCR amplification. So far, we did not conduct these experiments and analyses because we only have metagenomic sequencing data. We should conduct these analyses in the future to support our current data.

---

## [Decision Letter · Decision Letter 1]

13 Nov 2023

PONE-D-23-06645R1Metagenomic analyses of 7000 to 5500 years old coprolites excavated from the Torihama shell-mound site in the Japanese archipelagoPLOS ONE

Dear Dr. Inoue, Thank you for submitting your manuscript to PLOS ONE. The manuscript is now greatly improved, but there are still a few minor issues that are need to be addressed. Therefore, we invite you to submit a revised version (minor revision) of the manuscript that addresses the points raised during the review process.

We look forward to receiving your revised manuscript.

Kind regards,

Yanpeng Li, Ph.D.

Academic Editor

PLOS ONE

Reviewers' comments:

Reviewer's Responses to Questions

**Comments to the Author**

1. If the authors have adequately addressed your comments raised in a previous round of review and you feel that this manuscript is now acceptable for publication, you may indicate that here to bypass the “Comments to the Author” section, enter your conflict of interest statement in the “Confidential to Editor” section, and submit your "Accept" recommendation.

Reviewer #1: All comments have been addressed

Reviewer #2: (No Response)

2. Is the manuscript technically sound, and do the data support the conclusions?

Reviewer #1: Yes

Reviewer #2: Partly

3. Has the statistical analysis been performed appropriately and rigorously? 

Reviewer #1: Yes

Reviewer #2: N/A

4. Have the authors made all data underlying the findings in their manuscript fully available?

Reviewer #1: Yes

Reviewer #2: No

5. Is the manuscript presented in an intelligible fashion and written in standard English?

Reviewer #1: Yes

Reviewer #2: Yes

6. Review Comments to the Author

Reviewer #1: (No Response)

Reviewer #2: This is my 2nd time to review this paper. Again, this is an interesting work, but major concern remains. No human genome construction cannot convince the difficulty for recovering viral genomes that are much shorter – e.g., the work cited in this study (ref 13): https://doi.org/10.1038/s10038-020-00841-6.

Some specific comments:

Line 100: 2-5 mg or 2-5 g? It’s confusing later “0.08-0.1g” was described at Line 106.

Line 102: Will UV light damage DNA, thus impact mapping?

Line 107: Can you provide the numbers for DNA amounts?

Line 111: Excluding what size of fractions?

Line 139: Why not using the new ICTV taxonomy?

Line 190-180: Is this expected or surprising?

Line 193: highly, do you mean well preserved? Then how do you know they are preserved well or not?

Table 2 and 3: Coverage% - ‘%’ was missing in Table 2. What does ‘*’ mean? How much coverage is enough to support a presence of the tested viruses in your samples?

Line 229-230: If you map the modern human metagenomes to the giant viral genomes, won’t you also get the mapping like you showed in the tables?

7. PLOS authors have the option to publish the peer review history of their article (what does this mean?). If published, this will include your full peer review and any attached files.

Reviewer #1: No

Reviewer #2: **Yes: **Zhi-Ping Zhong

---

## [Author Response · Author response to Decision Letter 1]

30 Nov 2023

Response to Reviewers 

To Reviewer #2

This is my 2nd time to review this paper. Again, this is an interesting work, but major concern remains. No human genome construction cannot convince the difficulty for recovering viral genomes that are much shorter – e.g., the work cited in this study (ref 13): https://doi.org/10.1038/s10038-020-00841-6.

Reply:

We express our gratitude to the reviewer for providing numerous critical and valuable comments that significantly improve the manuscript. The manuscript has been thoroughly revised in accordance with the insightful suggestions.

We acknowledge the challenges associated with conducting viral discoveries using our highly fragmented DNA data. In this instance, we have presented the results of aligning these ultra-short reads to reference viral genomes, suggesting the potential existence of these viruses. It is crucial to emphasize that, as of now, we have not definitively identified specific viruses in our coprolite samples.

Some specific comments:

1. Line 100: 2-5 mg or 2-5 g? It’s confusing later “0.08-0.1g” was described at Line 106.

Reply:

2-5 g was accurate, and we have rectified the manuscript. We apologize for the error in writing.

2. Line 102: Will UV light damage DNA, thus impact mapping?

Reply:

The methodology is a standard for extracting ancient DNA. UV light has the potential to degrade and deactivate DNA, posing a risk of contamination on the sample surface. It is considered that the impact of UV light may not extend to ancient DNA located in the inner part of the samples. For example, Farrer et al. (2021) utilized dental calculus samples and evaluated various ancient DNA extraction protocols, including UV irradiation. Their findings revealed that both EDTA pre-digestion and a combined treatment involving UV irradiation and 5% sodium hypochlorite immersion were effective in reducing the proportion of environmental taxa and increased the abundance of oral taxa compared to untreated samples.

3.Line 107: Can you provide the numbers for DNA amounts?

Reply:

We revised the sentence and included the DNA amounts as follows: Four DNA extracts from TH55, TH58, TH62, and TH74 containing 5.80 ng, 4.46 ng, 15.54 ng, and 5.36 ng of DNA, respectively, were proceeded to library preparation for metagenomic sequencing.

4.Line 111: Excluding what size of fractions?

Reply:

Since we conducted size selection with x0.9 volume of AMPure XP Beads, libraries exceeding 250 bp (with an insert size of 130 bp) in total length were eliminated.

5. Line 139: Why not using the new ICTV taxonomy?

Reply:

Despite the recent abolition of the major families of Caudovirales last year, numerous virome studies continue to rely on the previous viral taxonomic classifications (e.g., Gregory et al. 2020). In order to maintain consistency with earlier reports, we have chosen to adhere to the old taxonomic classifications.

6. Line 190-180: Is this expected or surprising?

Reply:

While we are not entirely sure which specific sentence you are referring to, we believe you might be alluding to the sentence containing ‘Although sequence variations were not detectable, due to the low depth of alignments, there might be similar or closely related viral species with the present-day gut viruses as those in the gut of Jomon people.’ If so, it was not so surprised by the fact the presence of modern gut bacteria in the gut environment of ancient individuals (e.g., Wibowo et al. 2021, Rampelli et al. 2021.) These gut bacteria could potentially serve as hosts for phages, therefore similar modern gut viruses might exist in the guts of ancient people. Evidently, there is a possibility of distinct viruses existing in the gut of ancient individuals. Further studies will be helpful in uncovering and understanding these potential differences. 

7. Line 193: highly, do you mean well preserved? Then how do you know they are preserved well or not?

Reply:

That is right. We changed the word from ‘highly’ to ‘well.’ 

 To further examine and analyze the new coprolite samples, several methods can be employed for checking preservation conditions, including macroscopic identification of food remains, confirming the extracted DNA amount, and conducting DNA sequencing using a relatively low-throughput instrument such as the Illumina MiSeq platform. The microscopic identification findings could be combined with archaeological data pertaining to pollen or food items recovered from the excavation sites. In DNA sequencing data, the longer length and ancient DNA damage pattern observed in reads originating from the host human, food remnants, bacteria, and viruses could potentially act as indicators of favorable preservation conditions. Once the preservation condition of the DNA is confirmed through screening with the MiSeq platform, subsequent sequencing can be carried out using a high-throughput instrument like the NovaSeq platform to obtain a substantial volume of reads. 

8. Table 2 and 3: Coverage% - ‘%’ was missing in Table 2. What does ‘*’ mean? How much coverage is enough to support a presence of the tested viruses in your samples?

Reply:

Thank you for bringing this point to our attention. We have updated the column name from 'Coverage (1<x)' to 'Coverage% (1<x)' as suggested. Additionally, we have provided an explanation for '*' in the context of using deduplicated reads. In Table 2, where the units of length were absent, we have now included the unit 'bp' and rectified the typo from 'PMD' to 'PMDS.' We have also added the explanation for '*' in all supplementary tables. 

Regarding coverages, we were unable to specify the required coverage percentage, so we incorporated all available data. Given that previous studies have identified microbial sequences with at least a few mapped reads (e.g., Rampelli et al., 2021), we also presented viral species with a minimum of one read for PMDS ≥ 3.

9. Line 229-230: If you map the modern human metagenomes to the giant viral genomes, won’t you also get the mapping like you showed in the tables?

Reply:

We have acquired mapping results of the giant viral genomes based on modern feces metagenomic data (e.g., PRJEB13870) which the average alignment length was approximately 20 bp. It is noteworthy that modern human metagenomic data differs from ancient coprolites in that the DNA does not undergo postmortem degradation. Consequently, in the case of modern human metagenomic data, ultra-short reads with alignment lengths shorter than 30 bp should be excluded during the filtering process, as they are considerably shorter than human DNA. In contrast, in our coprolite samples, human DNA was highly degraded, and as a result, ultra-short reads were retained.

---

## [Editor Report · Decision Letter 2]

4 Dec 2023

Metagenomic analyses of 7000 to 5500 years old coprolites excavated from the Torihama shell-mound site in the Japanese archipelago

PONE-D-23-06645R2

Dear Dr. Inoue,

We’re pleased to inform you that your manuscript has been judged scientifically suitable for publication and will be formally accepted for publication once it meets all outstanding technical requirements.

Kind regards,

Yanpeng Li, Ph.D.

Academic Editor

PLOS ONE
---

## [Editor Report · Acceptance letter]

26 Dec 2023

PONE-D-23-06645R2 

PLOS ONE

Dear Dr. Inoue, 

I'm pleased to inform you that your manuscript has been deemed suitable for publication in PLOS ONE. Congratulations! Your manuscript is now being handed over to our production team.

Kind regards, 

on behalf of

Prof. Yanpeng Li 

Academic Editor

PLOS ONE